# Physical Activity, Dietary Behavior, and Body Weight Changes during the COVID-19 Nationwide Level 3 Alert in Taiwan: Results of a Taiwanese Online Survey

**DOI:** 10.3390/nu14224941

**Published:** 2022-11-21

**Authors:** Hao-Ting Ke, Chi-Lin Hsieh, Wei-Jen Cheng

**Affiliations:** 1Center for Traditional Chinese Medicine, Chang Gung Memorial Hospital, Taoyuan 333, Taiwan; 2Department of Internal Medicine, Chang Gung Memorial Hospital, Taoyuan 333, Taiwan; 3Graduate Institute of Clinical Medical Sciences, College of Medicine, Chang Gung University, Taoyuan 333, Taiwan; 4School of Traditional Chinese Medicine, Chang Gung University, Taoyuan 333, Taiwan

**Keywords:** COVID-19, physical activity, eating habits, lifestyle, bodyweight

## Abstract

This cross-sectional study aimed to explore the influence of the COVID-19 pandemic on physical activity (PA) and dietary habits, and their impact on body weight changes during the Level 3 alert period that resulted in the lockdown in Taiwan. The study was conducted between 1 July 2021 and 15 July 2021, using a Google Forms online survey platform. Personal data, anthropometric information, PA information, and dietary habit information were collected before and during the alert period. Exactly 374 respondents, aged between 20 and 66, were included in the study. The results indicate that the lockdown during the alert period negatively impacted all levels of PA, including vigorous and moderate activities and walking. Additionally, respondents showed a sedentary lifestyle, with an increased daily sitting time of 22%. However, body weight and dietary behavior were not significantly affected, and some dietary questions achieved significant differences, including eating three meals less regularly, among others. During the pandemic, exercise was still one of the most important ways to maintain health; therefore, we hope to bring more attention to the prevention of sedentary lifestyles and dietary abnormalities in Taiwan during a pandemic.

## 1. Introduction

Coronavirus disease 2019, or COVID-19, is a novel coronavirus caused by severe acute respiratory syndrome coronavirus 2 (SARS-CoV-2) [1,2]. Since the first case reported in Wuhan, China, in December 2019, COVID-19 has affected more than 60 countries worldwide, with over 6000 cases and 106 deaths, only two months after the outbreak [3]. Despite its proximity to and its dense economic activities with China, Taiwan took rapid responses and quickly instituted specific approaches for a containment strategy and resource allocation to control the outbreak [4,5,6]. Thus, there were only 443 cases and seven deaths as of 7 June 2020. However, on 19 May 2021, due to the surge in the number of indigenous cases and the increasing number of deaths, Taiwan’s Central Epidemic Command Center (CECC) raised the COVID-19 alert to Level 3 nationwide. The control measures of Level 3 alerts include mandatory wearing of masks in public; closing entertainment venues; banning physical activities in open spaces, gyms, and sports centers; and food vendors offering only takeout services [7]. While these restrictions helped to reduce the rate of infection, such limitations have led to numerous behavioral changes, such as social distancing and prolonged homestay, which caused negative effects by reducing physical activity (PA) and restricting access to many forms of exercise (e.g., closed civil sports centers and gyms, restrictions on walking distance, and insufficient space for exercise at home) [8,9,10]. Exercise is one of the most important risk factors for major disease morbidity. Multiple studies in humans and animals have demonstrated that regular exercise is positively linked to immune function [11]. Additionally, the benefits of exercise may reduce the COVID-19 fatality rate due to its positive effect on COVID-19-related comorbidities (e.g., cardiovascular disease, hypertension, type-2 diabetes, and obesity) [12]. 

In addition to limiting access to PA, banning indoor dining and limiting the number of people in supermarkets and grocery stores may change eating habits because it is less easy to obtain food than usual. Additionally, limited access to fresh food can have a negative effect on overall health both physically and mentally [13]. Combined with a structural barrier to maintaining a physically active lifestyle during the lockdown period, impaired nutritional habits could produce a state of positive energy balance, in which caloric intake exceeds energy expenditure, which can cause weight gain due to increased body fat [10]. Therefore, this study aimed to investigate the effect of the COVID-19 pandemic on physical activity and dietary behavior, and to further understand the relationship between physical activity, dietary behavior, and changes in body weight during the COVID-19 nationwide Level 3 alert period in Taiwan.

## 2. Materials and Methods

### 2.1. Study Design 

This cross-sectional study investigated the changes in physical activity and dietary habits, and their impact on body weight during the COVID-19 pandemic in Taiwan. 

### 2.2. Participants 

Participants were recruited through social media platforms, including Facebook^TM^ (Meta Platforms Inc., Menlo Park, CA, USA), WhatsApp^TM^ (Meta Platforms Inc.), Instagram^TM^ (Meta Platforms Inc.), and Line^TM^ Corporation, Tokyo, Japan). We included people older than 20 years who were living in Taiwan during the nationwide Level 3 COVID-19 alert period. All participants were fully informed about the study requirements, and that the results could be used for research purposes. Each participant could answer only once. They could withdraw their participation in the survey at any point before submission, and incomplete responses could not be saved. To maintain and protect the confidentiality of participants, their personal information and data were anonymized.

### 2.3. Survey Development and Privacy

The questionnaire was administered during the nationwide Level 3 COVID-19 alert period in Taiwan, using the Google Forms^®^ online survey platform. A link to the survey was shared on social media (Facebook^TM^, WhatsApp^TM^, Instagram^TM^, and Line^TM^). Participants completed the questionnaire directly connected to the Google platform, and the final database was downloaded as a Microsoft Excel spreadsheet. 

This study was approved by the Institutional Review Board (IRB) of the Chang Gung Medical Foundation. As participation was voluntary and survey responses were anonymous, the IRB waived the need for informed consent.

### 2.4. Survey Questionnaires 

The questionnaire included three sections with 46 questions: (1) personal data (including gender, age, level of education, marital status, and employment status) and anthropometric information (including height and weight); (2) physical activity information, using the Taiwanese version of the International Physical Activity Questionnaire Short Form (IPAQ-SF) [14]; and (3) dietary habit information, using the Diet Behavior Questionnaire from Health Promotion Administration, Ministry of Health and Welfare, Taiwan (HPA-MOHW) [15]. This was a self-report survey that asked participants to recall information regarding body weight, physical activity, and dietary habits before the Level 3 alert was launched (19 May 2021) and over the past 7 days. No actual measurements were taken. The entire questionnaire was completely anonymous and the respondents provided replies for both timeframes in one survey.

#### 2.4.1. Sociodemographic Characteristics 

Information on age, sex, level of education, marital status, employment status, height, and weight before and during the Level 3 alert was collected.

#### 2.4.2. Physical Activity Levels 

Changes in physical activity levels due to Level 3 alerts were identified using the Taiwanese version of the IPAQ-SF. The CVI of the IPAQ-Taiwan version was higher than 0.9, indicating good content validity. The retest reliability was 0.67 and the concurrent validity was 0.63, which was similar to other language versions, and it can be used as a tool to compare the physical activity levels of people around the world [16]. According to the official IPAQ-SF guidelines, data from the IPAQ-SF were summed within each category, including vigorous activity (V), moderate activity (M), and walking (W), to estimate the total amount of time engaged in PA per week [17,18]. Moderate activities are defined as walking with more than five kilograms of weight on one’s back (carrying children or heavy items), more laborious household tasks, tai chi, etc. Vigorous activities refer to weight training, hiking, long-distance running, playing ball, aerobic dance or riding a spinning bike, etc. The questionnaire referred to the PA of the last 7 days, and each activity had to take at least 10 min to be included. The respondents’ total weekly PA was estimated in metabolic equivalents of tasks performed in minutes per week (MET-min/week). This was calculated by multiplying three parameters: the MET value of an activity, the duration of the activity (in minutes), and the number of times the activity was carried out (V = 8 METs, M = 4 METs, W = 3.3 METs). We then added the MET minutes achieved in each category to obtain the total MET minutes of PA per week. The time spent sitting was also recorded in the questionnaire in order to assess sedentary behavior.

#### 2.4.3. Dietary Habits Information

The impact of Level 3 alerts on dietary habits was assessed using the Diet Behavior Questionnaire from the HPA-MOHW. This is an online survey sourced from “Overcoming Obesity: A Weight Loss Guide” published by the National Health Service, Ministry of Health and Welfare in 1997 [15]. The questionnaire had 12 questions, namely “Q1. Eating three meals regularly”, “Q2. No snacks or sweets”, “Q3. Eat slowly and chew carefully”, “Q4. No sugary drinks”, “Q5. Avoid high-fat foods”, “Q6. Eat fruits every day”, “Q7. Eat green vegetables every day”, “Q8. Eat late-night snacks”, “Q9. Eat while watching TV or reading books”, “Q10. Emotional eating behavior”, “Q11. Eat as a reward”, and “Q12. Eat only when hungry”. 

The response choices and the designated scores for the first seven questions were as follows: “Always” = 3; “Most of the time” = 2; “Sometimes” = 1; “Never” = 0. The response choices and the designated scores for the eighth to twelfth questions were as follows: “Always” = 0; “Most of the time” = 1; “Sometimes” = 2; “Never” = 3. The total score of this questionnaire corresponded to the sum of the scores in the 12 questions ranging from 0 to 36, where “0~12” represents very unhealthy dietary behavior, “13~20” represents fine dietary behavior, “21~30” represents good dietary behavior, and “31~36” represents very healthy dietary behavior.

### 2.5. Data Analysis 

Descriptive statistics were used to determine the demographic and personal characteristics and anthropometric parameters of the study sample. Data are represented as numbers and percentages (*n* (%)) for categorical variables or as mean and standard deviation (M ± SD) for continuous variables. To calculate the significant differences in responses before and during the Level 3 alert period, paired-sample *t*-tests were used. 

To determine the magnitude of the change in the score, Cohen’s d_s_ was calculated as the effect size for the comparison between two means. This was interpreted as 0.2 to be considered a “small” effect size, 0.5 for a “medium” effect size, and 0.8 for a “large” effect size. All of the statistical analyses were performed using GraphPad Prism 9.4.1 and Microsoft Excel 2019. The significance level was set at *p* < 0.05.

## 3. Results

### 3.1. General Characteristics of the Participants 

The nationwide Level 3 COVID-19 alert in Taiwan began on 19 May 2021, and lasted until 27 July 2021. The web survey was launched on 1 July 2021, and was concluded on 15 July 2021. In total, the data of 374 respondents (243 female and 131 male participants) aged between 20 and 66 years were analyzed, with four of the original 378 respondents excluded due to missing data. 

Overall, 65.0% of the participants were female. The general characteristics, including education level, marital status, and employment status, are presented in Table 1.

### 3.2. Anthropometrics Information before and during the Level 3 Alert Period

Table 2 presents the changes in body weight. The results indicated no significant differences in body weight before and during the Level 3 alert period.

### 3.3. Physical Activity Level Changes during the Level 3 Alert Period 

The responses to the physical activity questionnaire recorded before and during the Level 3 alerts are shown in Table 3. There was a significant decrease in all of the physical activity levels. The effect of the COVID-19 Level 3 alert on different gender and age groups for all of the PA MET values are shown in Table 4 and Table 5. All gender and age groups, except for 60–69 years, showed significant differences.

The number of times/week of vigorous activity decreased by 38.0% during the Level 3 alert, compared with before (*p* < 0.001, *d* = 0.409) and the number of minutes/time decreased by 48.7% (*p* < 0.001, *d* = 0.555). In addition, the MET values of vigorous activity were 47.5% lower during the Level 3 alert period compared with before (*p* < 0.001, *d* = 0.418). As for moderate activity, the number of times/week and minutes/time decreased by 17.6% (*p* < 0.001, *d* = 0.187) and 26.1% (*p* < 0.001, *d* = 0.275), respectively, compared with before. In addition, the MET values of the moderate activity were 27.0% lower during the Level 3 alerts (*p* < 0.001, *d* = 0.212). As for walking, the number of times/week and minutes/time during the Level 3 alerts, compared with before, decreased by 39.2% (*p* < 0.001, *d* = 0.713) and 37% (*p* < 0.001, *d* = 0.432), respectively. Additionally, the MET values of walking were 48.3% lower during the Level 3 alerts (*p* < 0.001, *d* = 0.476). Overall, the MET values of all PA were 44.1% lower during the Level 3 alerts (*p* < 0.001, *d* = 0.523). In terms of the effects of the COVID-19 Level 3 alert on different gender and age groups, male participants and those aged 20–29 years were impacted the most, as their PA MET values were 46.1% (*p* < 0.001, *d* = 0.606) and 48.4% (*p* < 0.001, *d* = 0.605) lower during the Level 3 alerts, respectively. Regarding the hours of sitting per day, the results increased by 22.2% during the Level 3 alert period (*p* < 0.001, *d* = -0.399).

### 3.4. Dietary Behavior Changes during the Level 3 Alert Period

#### 3.4.1. Total Score of Diet Behavior Questionnaire

With regard to the dietary behavior status, no significant difference was observed between the total scores of the dietary behavior results during the Level 3 alert compared to before (Table 6). The subgroups of the dietary behavior status before and during the Level 3 alert period are presented in Table 7.

#### 3.4.2. Score of Each Dietary Question

The recorded scores in each response to the dietary behavior questionnaire before and during the alert period are shown in Figure 1. The Q1 score (eat three meals regularly) decreased significantly during the period (*t* = 2.528, *p* = 0.0119, *d* = 0.17). The Q2 score (no snacks or sweets) increased significantly during the period (*t* = 3.054, *p* = 0.0024, *d* = −0.19), and the Q11 score (eat as reward) was significantly higher during the period (*t* = 2.192, *p* = 0.0290, *d* = −0.14). As for the remaining questions, no significant difference was found before and during the alert period.

## 4. Discussion

Different studies and surveys that have been conducted worldwide suggest that the COVID-19 lockdown affected the population’s daily lifestyle [19,20,21,22,23]. This cross-sectional study provides information regarding the impact of the COVID-19 lockdown on body weight, physical activity, and dietary behavior changes among Taiwanese residents during the Level 3 alert period. There were 374 replies, and the female respondents were about twice the number of males; 81.1% of the respondents were 20–49 years; 93.3% of the respondents were educated to college degree or above. According to a report on the age structure of the Taiwanese population in 2020 [24], the percentage of the population aged 20–49 years was about 52.5%. Statistics from Taiwan’s Ministry of the Interior, Department of Statistics showed that 47.3% of citizens aged over 15 years had a college degree or above by the end of 2020 [25]. Those aged 25–29 years had the highest percentage (81.42%) of people with a college degree or above, and 63.90% had a college degree. Our demographic data did show some bias toward younger populations and higher education levels. A review study comparing Facebook^TM^ recruitment of participants with traditional recruitment methods showed an over representation of females, young adults, and people with higher education levels and incomes [26], which was similar to our survey population. 

The findings from the online survey confirmed that the COVID-19 Level 3 alert negatively affected all levels of PA, including vigorous and moderate activity, and walking, as was initially expected. Furthermore, we found that respondents had a sedentary lifestyle with an increased daily sitting time of 22%. While the effect size for most values was small to medium, a 39.2% reduction in the number of times per week on walking, a 48.7% reduction in the number of minutes per week on vigorous activity, and a 44.1% reduction in MET minutes per week on overall PA were medium to large. This result is consistent with recent studies, indicating that COVID-19 home confinement could greatly impact activities in life, including participation in sports activities and PA engagement [19]. We also saw a 46.1% reduction and 48.4% reduction in MET values on overall PA in the male group and the group aged 20–29 years, respectively, showing that these groups were impacted the most. This result is consistent with a study from Bangladesh [27], which showed that the prevalence of physical inactivity during the COVID-19 pandemic was significantly higher among young adults compared with other groups. A possible explanation could be the closure of schools, which led to a lack of physical education courses, meaning that young adults were more prone to spend more time on the internet and electronic devices, leading to a decrease in all levels of PA. In terms of gender, it is traditionally believed that females tend to have more household chores compared with males. More time spent at home during the Level 3 alert could have led to more housework, which could be one possible reason why females showed less of a decrease in overall PA MET values than males. Yet, our results are inconsistent with those of previous studies conducted in Taiwan on investigating exercise behavior during the COVID-19 pandemic. In this study, the results showed that most Taiwanese maintained their exercise frequency, duration, or intensity during the COVID-19 pandemic, with only 20% of individuals decreasing their exercise frequency [28]. However, it is worth noting that the study was conducted from 7 April to 13 May 2020, and the COVID-19 Level 3 alert in Taiwan was not launched at that time. Although Taiwan did not have a large-scale lockdown compared with other parts of the world, Level 3 alert restrictions, such as shutting most business and public places, banning group gatherings and physical activities in open spaces, gyms, and sports centers, may still have created a barrier to maintaining exercise behavior among Taiwanese individuals [7]. While public life restrictions associated with the pandemic were usually of a limited duration, some studies have shown that even a short-term increase in physical inactivity and sedentary behavior can have significant adverse effects [29]. For example, 14 days of step reduction can lead to insulin resistance as well as increased central and liver fat and dyslipidemia, while reducing cardiorespiratory fitness [30]. In another study, 5 days of bed rest was seen to lead to a significant decrease in endothelial function, increased arterial stiffness, and elevated diastolic blood pressure [31]. Although most of these changes are reversible by a resumption of habitual physical activity in younger individuals, this is less true for the elderly and for individuals with metabolic disease [32]. Therefore, identifying ways to maintain or even enhance PA during the confinement periods seems to be of the utmost importance.

Before the COVID-19 pandemic, insufficient physical activity had already been recognized as a public health problem globally. Out of 168 countries, 27.5% of adults do not achieve the levels of physical activity necessary for good health [33]. Home confinement created a structural barrier to maintaining a physically active lifestyle; with a drastic drop in PA and exercise levels, while dietary behaviors remain unchanged or fail to counteract this inactivity lifestyle, leading to a positive energy balance that will gradually cause weight gain [10,34]. However, no significant changes in body weight were observed in our study, inconsistent with a previous study showing an increase in self-reported body weight among obese patients. Even when adjusted for physical activity, age, and gender, the habit of taking snacks remained a risk factor [35]. Additionally, another study showed weight gain between 0.5 and 1.8 kg (±2.8 kg) after just 2 months of quarantine, and risk factors included increased eating in response to sight and smell, emotional eating, and increased snacking frequency [36,37]. Contrary to our expectations, there was no significant difference between the total scores of dietary behavior during and before the COVID-19 Level 3 alert period in our study. However, some dietary questions achieved a significant difference, including eating three meals less regularly, eating fewer snacks or sweets, and being less likely to eat as a reward. The negative changes in regular eating can be attributed to remote work from home and distance learning. When work and study activities were shifted to home, the lack of a clear routine of time and space caused a free work schedule, thus leading to irregular meals. Some good eating habits, including eating fewer snacks or sweets and being less likely to eat as a reward, were significantly increased during this period. A possible explanation for this could be the differences in living habits between Taiwan and Western countries. Owing to the popularity of the Internet, consumer electronics (cell phones, TVs, PCs, play devices, etc.), developed delivery services, and online shopping in Taiwan, it is easy to purchase food and daily necessities, thus reducing anxiety and stress from the shortage of supplies. In addition, Taiwanese people tend to believe that one should eat healthier before and after receiving COVID vaccines. A healthy diet prevents hyper-inflammation and might help to protect against inflammation from virus infections [38]. The concept of a health regimen based on traditional Chinese medicine is deeply rooted in the minds of the Taiwanese people, which could be a possible explanation for why they consumed fewer snacks during the COVID pandemic.

Our study had several limitations. First, the sample size was small and mostly comprised of females, aged 20–59 years, and individuals with a higher education levels, which may not be representative of the population. Second, the data were collected through a web-based survey, which might have limited the participation of certain groups (e.g., groups that do not have Internet access or who do not use the Internet often). However, the Internet use rate among Taiwanese people aged 12 years and above reached 83.0% in 2020 [39], suggesting high Internet usage by the Taiwanese. Moreover, the Internet use rate was over 90% among those aged below 55 years, 86.2% among those aged 55–59 years, and 72.8% among those aged 60–64 years, but only 42.7% among those aged 65 years and above, which could explain our respondents’ age distribution. Third, because of the cross-sectional design of the study and the use of a self-report questionnaire, the existence of recall bias cannot be ruled out, which may have resulted in an over or underestimation of the results. However, the IPAQ is an international tool for the assessment of PA and time spent sitting, and is widely used because of its simplicity and inexpensiveness, and its validity and reliability have been proven in numerous studies [40,41,42]. Fourth, we asked participants to recall their body weight, physical activity, and dietary habits before the Level 3 alert was launched (19 May 2021); however, we did not clarify the length of the recall period, meaning that it could be any amount of time before the alert. Additionally, the body weight was self-reported, so the data might not be accurate. However, it is very common for patients in Taiwan to measure their blood pressure as well as height and weight before visiting a clinic or hospital, and it is also common to have a weight scale at home. Therefore, it should not be too difficult to measure one’s body weight at home.

## 5. Conclusions

The results of this cross-sectional study indicate that the lockdown period during the COVID-19 Level 3 alert had a negative impact on all PA levels of Taiwanese adults, with a significant increase in sitting time, leading to a more sedentary lifestyle. However, body weight and dietary behavior were not significantly affected, probably because the changes in PA level could not be reflected in weight changes over such a short period. Furthermore, the body composition was not measured; therefore, we could not determine the relative variation in total fat and skeletal muscle mass proportion. In addition, eating three meals irregularly could be attributed to remote work from home and distance learning during the Level 3 alert period, which is a bad eating habit that can easily lead to obesity. Nevertheless, some good eating habits, including eating fewer snacks or sweets and being less likely to eat as a reward, were significantly increased during this period. This may also explain the lack of significant changes in body weight during the lockdown period.

During the pandemic, exercise is still one of the most important ways to effectively maintain health [2]. Government health organizations should be aware that reducing the possibility of performing PA during Level 3 alerts might have negative consequences on the health status of the population. According to the American College of Sports Medicine, 150–300 min of moderate-intensity aerobic exercise and two sessions of resistance exercise per week are recommended to maintain a healthy body [43]. Although adverse effects on short-term physical inactivity are reversible by habitual PA in younger individuals, this is less true for the elderly. Therefore, it is important to identify ways to enhance PA during confinement periods.

## Figures and Tables

**Figure 1 nutrients-14-04941-f001:**
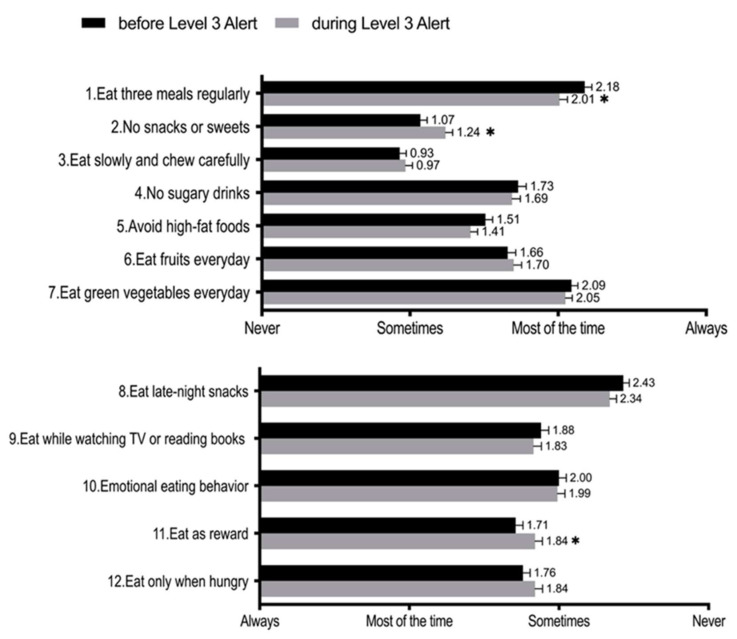
Participants’ scores in response to the related diet behavior questions. * Significant differences between “before” and “during” the COVID-19 Level 3 alert period.

**Table 1 nutrients-14-04941-t001:** Participant characteristics.

		*n*	%
Gender			
	Male	131	35.0%
	Female	243	65.0%
Age (years)	20–29	175	46.8%
	30–39	56	15.0%
	40–49	72	19.3%
	50–59	55	14.7%
	60–69	16	4.3%
Level of education			
	Graduate school or above	83	22.2%
	University	266	71.1%
	Senior/vocational high school	23	6.1%
	Junior high school or below	2	0.5%
Marital status			
	Single	171	45.7%
	Married/partnership	195	52.1%
	Divorced/widowed	8	2.1%
Employment status			
	Employed	244	65.2%
	Self-employed	27	7.2%
	Unemployed	27	7.2%
	Student	58	15.5%
	Retired	17	4.5%
	Unable to work	1	0.3%

**Table 2 nutrients-14-04941-t002:** Body weight changes before and during the Level 3 alert period.

	Before Level 3 Alert	During Level 3 Alert	*n*	Difference (%)	Student *t*	
					*T*	*p*
Female	58.04 ± 11.10	58.15 ± 11.13	243	0.12 (0.2%)	0.9794	0.3284
Male	73.95 ± 11.41	73.89 ± 11.26	131	−0.06 (−0.08%)	0.2504	0.8027

**Table 3 nutrients-14-04941-t003:** Effect of the COVID-19 Level 3 alert on the physical activity level.

		Before Level 3 Alert (*n* = 374)	During Level 3 Alert (*n* = 374)	Difference (%)	Student *t*		Cohen’s d_s_
					*t*	*p*	
Vigorous activity (V)	times/week	1.63 ± 1.54	1.01 ± 1.52	−0.62 (−38.0%)	8.23	<0.001	0.409
	mins/time	37.66 ± 36.47	19.33 ± 29.25	−18.33 (−48.7%)	10.47	<0.001	0.555
	MET values	780.32 ± 1008.97	410.05 ± 744.12	−370.3 (−47.5%)	8.05	<0.001	0.418
Moderate activity (M)	times/week	1.59 ± 1.53	1.30 ± 1.47	−0.28 (−17.6%)	3.92	<0.001	0.187
	mins/time	30.28 ± 31.15	22.38 ± 26.19	−7.90 (−26.1%)	5.07	<0.001	0.275
	MET values	290.21 ± 401.87	211.60 ± 339.71	−78.61 (−27.0%)	4.67	<0.001	0.212
Walking (W)	times/week	3.32 ± 1.80	2.02 ± 1.85	−1.30 (−39.2%)	14.18	<0.001	0.713
	mins/time	40.99 ± 37.64	25.83 ± 32.48	−15.16 (−37.0%)	8.71	<0.001	0.432
	MET values	555.49 ± 624.22	286.94 ± 498.24	−268.5 (−48.3%)	9.59	<0.001	0.476
All physical activities	MET values	1626.02 ± 1527.25	908.60 ± 1198.23	−717.4 (−44.1%)	10.80	<0.001	0.523
Sitting	hours/day	5.8 ± 3.0	7.1 ± 3.5	1.29 (22.2%)	6.82	<0.001	−0.399

**Table 4 nutrients-14-04941-t004:** Effects of COVID-19 Level 3 alert on gender for all of the PA MET values.

All PA MET Values						
Gender	Before Level 3 Alert	During Level 3 Alert	Difference (%)	Student *t*		Cohen’s d_s_
				*t*	*p*	
Female (*n* = 243)	1313.12 ± 1237.23	756.95 ± 953.36	−556.2 (−42.4%)	7.71	<0.001	0.505
Male (*n* = 131)	2206.43 ± 1822.22	1189.90 ± 1518.16	−1017.0 (−46.1%)	7.78	<0.001	0.606

**Table 5 nutrients-14-04941-t005:** Effects of the COVID-19 Level 3 alert on different age groups for all of the PA MET values.

All PA MET Values						
Age	Before Level 3 Alert	During Level 3 Alert	Difference (%)	Student *t*		Cohen’s d_s_
				*t*	*p*	
20–29 y (*n* = 175)	1876.02 ± 1719.73	967.84 ± 1241.66	−908.2 (−48.4%)	8.41	<0.001	0.605
30–39 y (*n* = 56)	1297.47 ± 1197.62	820.31 ± 1316.14	−477.2 (−36.8%)	2.84	0.0062	0.380
40–49 y (*n* = 72)	1314.69 ± 1120.53	733.63 ± 1055.70	−581.1 (−44.2%)	5.02	<0.001	0.536
50–59 y (*n* = 55)	1550.75 ± 1652.85	923.48 ± 1167.27	−627.3 (−40.4%)	3.76	<0.001	0.441
60–69 y (*n* = 16)	1680.47 ± 1047.75	1291.78 ± 975.70	−388.7 (−20.1%)	1.99	0.0642	0.397

**Table 6 nutrients-14-04941-t006:** Participants’ total scores of dietary behavior before and during the Level 3 alert period.

	Total Scores	Difference (%)	Student *t*	
			*t*	*p*
Before Level 3 alert	20.94 ± 5.42	−0.02 (0.1%)	0.657	0.5
During Level 3 alert	20.92 ± 5.76			

**Table 7 nutrients-14-04941-t007:** Subgroups of dietary behavior status before and during the Level 3 alert period.

Total Scores (Dietary Behavior Status)	Before Level 3 Alert	During Level 3 Alert
0~12 (Very unhealthy)	23	31
13~20 (Fine)	147	152
21~30 (Good)	190	170
31~36 (Very healthy)	14	21

“0~12” represents a very unhealthy dietary behavior, “13~20” represents a fine dietary behavior, “21~30” represents a good dietary behavior, and “31~36” represents a very healthy dietary behavior.

## Data Availability

All of the experimental data are included in the figures and manuscript. The data presented in this study are available upon request from the corresponding author.

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
