# Peer review of "Physical Activity, Dietary Behavior, and Body Weight Changes during the COVID-19 Nationwide Level 3 Alert in Taiwan: Results of a Taiwanese Online Survey"

_nutrients, 2022, doi:10.3390/nu14224941_

Round 1

Reviewer 1 Report

The study described an online survey result of physical activity, dietary behavior, and body weight changes during COVID-19 Nationwide lockdown in Taiwan. Some of the results would be expected because many similar studies from other countries, and some were different. However, the lockdown period is also short. It would be great if the author could talk about short term effect of physical inactivity and dietary changes. Also noted that the reference and previous study was based on long-term physical inactivity and its non-cause effective association with COVID. It is important to be cautions to make statement and compare to other studies to realize if it is an appropriate comparison. Please double check the statements in line 46-47, line 219, line 235-237 ;line 249-251 and make necessary adjustments. 

In the methods, did you mention how many people you reached through how many platforms? I wonder about the response rate.

In results, it is good that you noticed the majority were female and young adults. Can you further breakdown the difference of results by age group or by gender to further explore the data set to see which age group actually has more impact by covid lockdown?

Reviewer 2 Report

I have found this manuscript to be well written and an interesting read, although the time of Covid-19 lockdowns are mainly behind us.

Below I have put some comments and questions that I feel need to be addressed before this can be published.

Lines 75 and 82 (about anonymity) are repetitive and need to be amended.

 Line 94 you have said no other personal information were requested but you have collected gender age and anthropometric data etc, which counts as personal data.  You just have not recorded their names.  Please reword line 94 to say that it was anonymous rather than no personal data

2.4.1  please state that this was self reported information and that no actual measurements were taken. It would be helpful to give more information on how body mass was recorded.  Did every participant have a set of weight scales?  It is very precise for everyone to know their pre lockdown weight.  How would they know this? Maybe in Taiwan it is part of the health system, but in the UK, people would not know their body mass unless they had scales at home and then we would  not know how acurate these are.  I think this also needs to go in the limitations.

 Need to reference the IPAQ-SF and the HPA-MOHW.

As so many of the participants were educated to university level, it would be good to state the percentage of the Taiwanese population who are educated to this level to see if it is a fair reflection of society in general of if it is a biased population in the study.

Tables 2 and 3.  It would be better to call ‘weight’, ‘mass’.  As there were no changes in table 2, table 3 is not needed.  Or just use table 3 and not table 2.

 Need to give a better explanation as to why people ate less snacks, although you have compared this to other studies, it needs a better explanation as to why you think your cohort ate less snacks. Did their eating redcue overall?  If there is no change in body mass or BMI are you sure that there was a change in snacks eaten?

 Can you explain how they are doing vigorous exercise if all outdoor physical activity is banned?  Would this be in their houses? 

Reviewer 3 Report

The paper presents a succinct report of an online survey regarding the impact of a Level 3 lockdown in Taiwan on reported physical activity, dietary habits, and body weight changes for adults. While the methodology seems straightforward, more clarity regarding the following would be helpful:

1) What was the actual timeframe for the "before" and "during" periods and how was this data collected? Did respondents report for both timeframes in one survey or did they complete an initial survey before the Level 3 lockdown and then a follow-up survey was sent? What then were the limitations regarding how the "before" and "during" data were collected?

2) Several survey instruments were included as part of the survey questionnaire. While it was reported that both the Taiwanese version of the IPAQ-SF and the Diet Behavior Questionnaire from HP-MOHW were previously validated, were these surveys specifically validated for use in an online survey and can this be specified in the paper?

Regarding the results and discussion, specific comments are:

3) The demographics were reported as adults ranging in age from 20-69 years, yet interestingly 60% were <40 years old and 93% had a university or graduate degree, and only 12% were unable to work/retired/unemployed. How do these demographics compare to the general adult population in Taiwan? How could a relatively young, educated, employed sample have influenced your results and how does the survey population compare to other studies?

4) Table 7 and Figure 1 present related and similar results. Suggest including either Table 7 or Figure 1, but not both, to avoid repetition.

Round 2

Reviewer 1 Report

Thank you for making all the changes to improve this manuscripts